

# Gpr149 is involved in energy homeostasis in the male mouse

Steven Wyler, Surbhi, Newton Cao, Warda Merchant, Angie Bookout and Laurent Gautron

Internal Medicine and Center for Hypothalamic Research, UT Southwestern Medical Center, Dallas, TX, United States

## ABSTRACT

GPR149 is an orphan receptor about which little is known. Accordingly, in the present study, we mapped the tissue expression of *Gpr149* in mice using three complementary approaches: quantitative PCR, *in situ* hybridization, and a newly generated Gpr149-Cre reporter mouse model. The strongest expressions of *Gpr149* were observed in neurons of the islands of Calleja, the ventromedial hypothalamus, and the rostral interpeduncular nucleus. Moderate-to-low expression was also observed in the basal forebrain, striatum, hypothalamus, brainstem, and spinal cord. Some *Gpr149* expression was also detected in the primary afferent neurons, enteric neurons, and pituitary endocrine cells. This expression pattern is consistent with the involvement of GPR149 signaling in the regulation of energy balance. To explore the physiological function of GPR149 *in vivo*, we used CRISPR-Cas9 to generate a global knockout allele with mice lacking *Gpr149* exon 1. Preliminary metabolic findings indicated that $Gpr149^{-/-}$ mice partially resist weight gain when fed with a high-fat diet and have greater sensitivity to insulin than control mice. In summary, our data may serve as a resource for future *in vivo* studies on GPR149 in the context of diet-induced obesity.

## INTRODUCTION

GPR149 is an orphan receptor about which very little is known. Prediction algorithms suggest it is a 732 amino acid (aa), Class-A, rhodopsin like, G-protein coupled receptor (GPCR) with clear orthologs found in many vertebrates including mice and humans (*Friedel et al., 2001*; *Edson, Lin & Matzuk, 2010*). Uniquely, GPR149 has a highly conserved 360 aa C-terminal domain with no homology to other proteins and that may play a role in downstream signaling. However, its signaling pathway and endogenous ligands are currently unidentified. Based on phylogenic analysis of all human GPCRs, GPR149 shows the closest sequence homology to those that use peptides as their ligand (*Vassilatis et al., 2003*). The first study entirely focusing on GPR149 biology was published in 2010 (*Edson, Lin & Matzuk, 2010*). In a later study, the same authors identified *Gpr149* as an oocyte-enriched gene using microarrays in mice and confirmed the expression for *Gpr149* mRNA in the ovaries (*Edson, Lin & Matzuk, 2010*). The intestines and brain were also shown to express *Gpr149* mRNA. The authors also generated a *Gpr149* knockout

Corresponding authors
Steven Wyler,
Steven.Wyler@UTSouthwestern.edu
Laurent Gautron,
laurent.gautron@utsouthwestern.edu

mouse to determine the necessity of GPR149 in reproductive functions, finding that GPR149-deficient females are hyperfertile.

Other studies have revealed that *Gpr149* expression in brain sites involved in energy balance and glucose homeostasis including, most notably, the ventromedial hypothalamus (*Ehrlich et al., 2018*; *Affinati et al., 2021*). Furthermore, quantitative PCR (qPCR) data showed that *Gpr149* is widely expressed in mouse brain regions, as well as in the eyes and pituitary gland (*Ehrlich et al., 2018*; *Affinati et al., 2021*). Recently, we also found that *Gpr149* is highly enriched in the mouse vagal afferents (*Egerod et al., 2018*). Another recent study also identified *GPR149* as a negative regulator of myelinization (*Suo et al., 2022*). However, the identity and distribution of *Gpr149*-expressing cells across the body remains largely unknown. Furthermore, its roles in modulating metabolic functions remain unexplored.

Accordingly, the present study was designed to serve as a resource for future *in vivo* studies on GPR149 by achieving two main goals: (1) establishing a comprehensive mapping of *Gpr149* distribution in a mouse model using qPCR and *in situ* hybridization (ISH); and (2) generating and validating novel transgenic mice useful for *in vivo* studies; specifically, a Gpr149-Cre line and a global knockout for *Gpr149*. Moreover, we have made steps towards characterizing the metabolic profile of *Gpr149*-deficient mice.

## MATERIALS AND METHODS

### Generation of novel transgenic lines and mouse care

Tomato reporter mice (stock #007909) and male C57BL/6J mice 6–8 weeks of age were purchased from Jackson Laboratory. All studies were approved by the University of Texas (UT) Southwestern's Institutional Animal Care and Use Committee (IACUC; APN# 2016-101590) and all mice were maintained in a barrier facility with controlled temperature (23 °C) on a 12 h light/dark cycle (lights on 0600–1800). Unless indicated otherwise for the purpose of a feeding study, animals were grouped housed and had unrestricted access to food and water. For enrichment and thermal comfort, nestlets and igloos were provided to all mice. Of note, euthanasia of animals strictly followed a protocol approved by our IACUC and the American Veterinary Medical Association. Animals with signs of visible distress, labored breathing, or excess weight loss (20% of initial weight) euthanized prior to the end of the experiment. Euthanasia of animals not needed for experiments was performed by carbon dioxide immediately followed by cervical dislocation. Animals needed for experiments were euthanized as described below (qPCR and tissue preparation sections). To the further extent possible, our manuscript followed the ARRIVE guidelines (Animal Research: Reporting of *In Vivo* Experiments) (*Percie du Sert et al., 2020*).

Transgenic mice were generated as follows: For the Gpr149 null allele: guides flanking a 2 kb region containing Gpr149 exon 1 were ordered from IDT: 5′ Guide ribonucleotides sequence: 5′-CUUAUAACUGGUCACCUAUGUGUUUUAGAGCUAUGCU-3′ 3′ ribonucleotide sequence: 5′-UUGGUAGUUAACGAGACCCCGUUUUAGAGCUAUGCU-3′. Guides, trcRNA, Cas9 protein were administered through a pronuclear injection in C57Bl/6N (Charles River) by the UT Southwestern Transgenic Technology Center. Founders were screened by PCR and

sanger sequencing. One line generated through non-homologous end joining was selected and expanded by crossing with C57Bl/6N mice from Charles River. Mice were genotyped using the following primers: Gpr149 1: 5′-GCTGCTTGTAATGTGTGCAGAGAG-3′ Gpr149 2: 5′-GTCTACTCATGGCAGACCAAAGTAATGG-3′ Gpr149 3: 5′-GTCTCTTGGTGCTAGAGATGGGTG-3′ resulting in a WT band of 200 bp and a Gpr149 KO band of 350 bp. For the Cre-P2A-Gpr149 mouse, the endogenous Gpr149 locus was targeted with the following guide ordered from IDT: 5′-AAGUCAUAAUUCUACGGAGAGUUUUAGAGCUAUGCU-3′. Pronuclei were coinjected with an IDT Megamer TM (Integrated DNA Technologies) encoding 100 5′ arm 84 3′ homology arms and the 1,119 bp Cre-P2A at Gpr149's ATG start sequence. Synonymous mutations were placed in the guide sequence to prevent recutting of the Cre-P2A insert. Sequences are also included in Table S1.

## Quantitative PCR (qPCR)

On the day of sacrifice, mice were anesthetized with an overdose of chloral hydrate (500 mg/kg, i.p.), followed by decapitation. Tissue samples were collected according to methods detailed in our previously published studies (*Bookout et al., 2006a, 2006b*). Total RNA was isolated using the Quick-RNA microprep Kit using manufacturer's instructions (Zymo, Irvine, CA, USA). cDNA was synthesized using the High-Capacity cDNA synthesis Kit following the manufacturer's protocol (Applied Biosystems, Waltham, MA, USA). cDNA levels were measured in triplicates using a QuantStudio 5 Real-Time PCR System (Applied Biosystems by Thermo Fisher Scientific, Foster City, CA, USA). Pre-validated Taqman assays for Gpr149 (Mm00805216_m1), and 18s (Hs99999901_s1), were purchased from Thermo Fisher Scientific. The relative amount of transcript levels was calculated using the delta/delta CT method.

## Tissue preparation

On the day of sacrifice, mice were anesthetized with an overdose of chloral hydrate (500 mg/kg, i.p.) and perfused transcardially with 0.9% saline followed by 10% neutral buffered formalin (Sigma–Aldrich, St. Louis, MO, USA). Brains and peripheral tissues were dissected and post-fixed in formalin for 24 h at 4 °C. Then, samples were incubated for 24 h at 4 °C in 20% sucrose made in 0.1 M phosphate-buffered saline (PBS). Free-floating coronal brain sections of 25 μm thickness were produced using a freezing microtome (Leica, Teaneck, NJ, USA), collected in PBS, and stored in a cryoprotectant at −20 °C. Sections from other tissues with a thickness of 14 μm were produced using a cryostat (Leica, Teaneck, NJ, USA), collected onto SuperFrost slides, and stored at −80 °C. For imaging native tdTomato, brain sections were rinsed in PBS, mounted onto SuperFrost slides, and coverslips with ProLong™ Gold Antifade (Thermo Fisher, Waltham, MA, USA) was placed on the samples. A total of three Gpr149-Cre-tdTomato mice were used to survey native tdTomato brains and peripheral organs. For other applications, tissues were processed as explained below.

## RNAScope *in situ* hybridization (ISH)

Free-floating brain sections were rinsed in PBS, incubated in a solution of $H_2O_2$ for 10 min, rinsed with PBS, carefully mounted on SuperFrost slides, and desiccated overnight at room temperature. The brain slides were then processed using a multiplex fluorescent kit (cat# 323110) or a chomogenic FastRed kit (Cat#322350) following the manufacturer's instructions. The $H_2O_2$ step in the ACD protocol was omitted because it was performed the day before. The *Gpr149* probe (cat# 318071) was applied at 40 °C for 2 h. Amplification steps were carried out using either Opal570 (cat# FP1488001KT; Akoya Biosciences, Japan) for fluorescence assays or FastRed (ACD) for chromogenic assays. Fluorescently labeled sections were counterstained with DAPI and mounted using ProLong medium. Chromogenically-labeled sections were counterstained with hematoxylin and mounted using Ecomount medium (Biocare medical, Concord, CA, USA). Fluorescent assays were applied on three entire brains from three C57BL/6J mice and three Gpr149-Cre-tdTomato mice. For validation, chromogenic assays were applied to the brain nodose ganglion, and pituitary gland of one $Gpr149^{-/-}$ and one $Gpr149^{+/+}$ mouse. When fluorescent RNAScope analysis was performed on the brain sections of *Gpr149*-Cre-tdTomato mice, the same procedure was followed. Native tomato was bright enough to be seen without any obvious decline in fluorescence intensity. Tissues other than the brain samples already collected on slides were processed in the same manner. However, the $H_2O_2$ step was performed on the hybridization day rather than the day before.

## Metabolic profiling

We assessed the long-term effects on body weight, food intake, and glucose homeostasis in cohorts of male mice maintained on either chow (No. 2016 Harlan, Teklad) or high fat diet (60% Fat; Research Diets D12492). We used a group size of 6–9 mice/group. Body weight for the chow mice was assessed weekly until week 13 and for the HFD until week 20. At the end of 20 weeks, body composition was analyzed by magnetic-resonance whole-body composition analyzer (EchoMRI). Upon the completion of the studies, tissues including liver, white and brown fat pads, heart, and skeletal muscles were weighted. Food intake was assessed in a cohort of fasted (16 h) mice. Upon refeeding, pellets of chow diets were weighted over a period of 4 h. A cohort was also used to assess daily food intake under *ad libitum* food access. Pellets of chow diets were weighted once a day over a period of four consecutive days (see Fig. S4 for diet composition). A separate cohort was used for insulin and glucose tolerance tests (ITTs and GTTs) on chow diet. Single housed mice were fasted 6 h prior to testing starting at 8:00 AM with water provided *ad libitum*. For the GTT mice received an intraperitoneal injection of glucose at 1 g/kg. For ITT, mice received 1 U/kg. Blood glucose from the tail vein was measured at five time points 0, 15, 30, 60, and 120 min using a handheld commercial glucometer (Bayer's Contour Blood Glucose Monitoring System; Bayer, Leverkusen, Germany).

## Imaging

Fluorescence-labeled whole brains sections (RNAscope or tdTomato) were scanned by the Whole Brain Microscopy Facility at UTSouthwestern (see Acknowledgments) using a

Zeiss Axioscan. Z1 and appropriate filters. We also used a confocal microscope (Zeiss LSM880) available at the Quantitative Light Microscopy Core (see acknowledgments) to image fluorescence-labeled tissues at higher magnifications. ImageJ (NIH, Bethesda, MD, USA) and Adobe Photoshop 2021 were used to uniformly adjust the resolution and contrast of all our digital images. Estimates of signal strengths were done by visually inspecting brain sections under epifluorescence microscopy (Leica, Teaneck, NJ, USA). Brain sites were visually identified and ranked by expression level with reference to the Franklin and Paxinos atlas (3rd edition).

## Data analysis

Quantitative data were presented as mean ±SEM. Differences between genotypes were compared with either a one-way (qPCR and physiological data) or two-way ANOVA analysis followed by the *post-hoc* Dunnett's test (body weights curves). Statistical significance is accepted at a value of $p < 0.05$. The exact values are indicated above bar graphs. Statistical tests and their tabular results (degree of freedom, effect size, *etc.*) are included in Supplementary figures for all test with statistically significant results. Graphs of numerical data were produced using GraphPad Prism 9. Groups sizes are included either in our results section, or figure legends, or graphs. For statistical analysis, mice carrying wild-type alleles were considered as control groups.

## RESULTS

### Mapping of Gp149-expressing sites reveals neuronal enrichment

We mapped the expression of *Gpr149* in mouse tissues by qPCR (Fig. 1). The strongest expression of *Gpr149* is observed in the central nervous system (CNS) tissues including, among other examples, the striatum, hypothalamus, brainstem, and spinal cord (Fig. 1). The highest *Gpr149* expression in a non-neuronal tissue is that in the pituitary gland (Fig. 1). Low levels of expression are observed for the gastrointestinal tract and female reproductive organs (Fig. 1). Other examined tissues showed very low or close-to-detection threshold expression. Raw data are included in Fig. S1.

The exact distribution of *Gpr149* within the CNS was further explored by fluorescent RNAscope ISH. At least 80 brain regions express *Gpr149* at varying levels. Figure 2 shows representative coronal brain sections with the highest *Gpr149* expression and Table 1 summarizes the relative expression levels across the brain. The strongest expression of *Gpr149* is observed in the islands of Calleja and surrounding nuclei, such as the olfactory tubercle, the ventromedial hypothalamus, the rostral interpeduncular nucleus, and a few select brainstem nuclei such as the sphenoid nucleus (Fig. 2; Table 1). Moderate-to-low expression is also shown in many brain sites across the basal forebrain, striatum, hypothalamus, brainstem, and spinal cord (Fig. 2; Table 1). Only low expression is observed in the cortical and subcortical regions such as the hippocampus. *Gpr149* expression appears to be restricted to neurons rather than glia in the adult brain. In fact, white matter tracts, meninges, and epithelia presented no discernable signals. The above roadmap of *Gpr149*-expressing sites in the mouse brain is largely consistent with Gpr149 playing a role in regulating basic motivated behaviors, autonomic outflow, and sensory

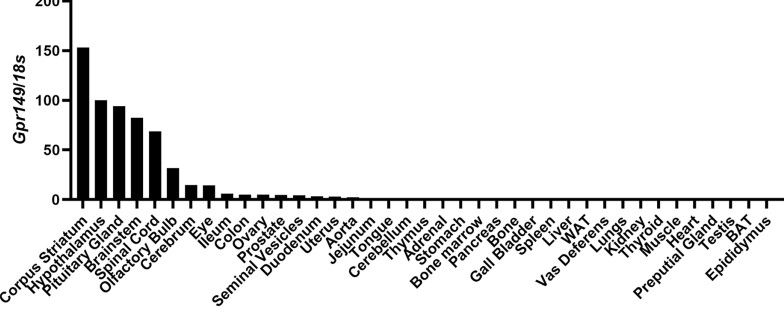

**TISSUE PANEL**

**Figure 1 QPCR analysis results for *Gpr149* in representative mouse tissues.** Highest *Gpr149* expression (normalized to 18s ribosomal RNA) is observed in the CNS and pituitary gland. Data were generated using the RNA pooled from three mice. The hypothalamus was used to normalize raw data.

processes (*Swanson, 2005*). As shown later, probe specificity was validated in a novel knockout model.

Cre mouse lines are useful for a wide range of applications, including the manipulation and mapping of cell types (*Madisen et al., 2010*). Here, using CRISPR-Cas9, we generated a Gpr149-Cre-expressing mouse by knocking an Cre-P2A sequence into the 5′ UTR of *Gpr149* (Fig. 3). This will allow for the coexpression of Cre and Gpr149 from the endogenous Gpr149 locus. Crossing this mouse to tdTomato reporter mice allowed us to further assess the expression of *Gpr149*. As a result, *Gpr149*-Cre-tdTomato mice present fluorescence across the entire brain following a pattern highly reminiscent of that of endogenous *Gpr149* mRNA (Fig. 4A). Cells resembling neurons and their axons are observed across the entire brain including, among other examples, the striatum (Fig. 4B). We also noticed several brain regions that contained brightly labeled axonal fibers rather than cell bodies. These include the substantia nigra (Fig. 4D) and primary sensory areas of the brainstem (Fig. 4E). Lastly, tdTomato-labeled cells across the brain always resemble neurons, except for cells lining the aqueduct (Fig. 4F). These are the only non-neuronal cells observed.

We further validated the specificity and sensitivity of this new Cre line by assessing tdTomato in combination with RNAScope for *Gpr149*. As anticipated, the *Gpr149* signals coincide very well with tdTomato-positive cell bodies across the brain. For example, in the basal forebrain around the island of Calleja, a site of high *Gpr149* mRNA expression, tdTomato fluorescence is very intense (Figs. 5A and 5B). As another example, in the striatum and cortex tdTomato-positive neurons always express *Gpr149* mRNA (Figs. 5C and 5D). However, the intensity of tdTomato fluorescence is not always correlated with the strength of mRNA expression. For instance, in the VMH, another site of high *Gpr149* mRNA expression, the neurons are only faintly fluorescent for tdTomato (Figs. 5E and 5F). Coincidence between tdTomato and *Gpr149* is also observed in the rest of the brain including the midbrain and brainstem (Figs. 5G and 5H). The only site with an apparent mismatch between tdTomato and *Gpr149* is the cerebral aqueduct. Tomato-positive cells lining the aqueduct either present either low levels or undetectable *Gpr149* signals

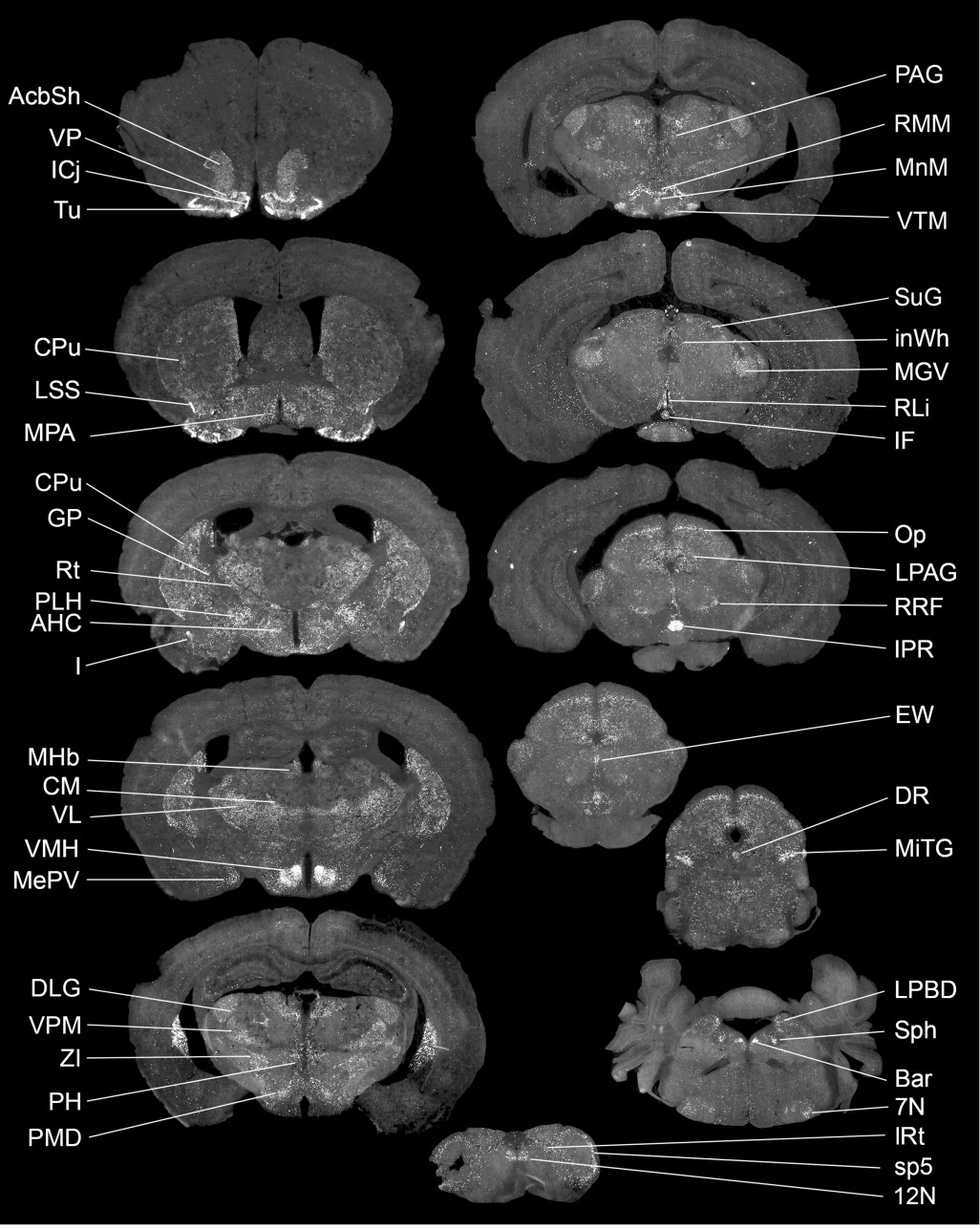

**Figure 2 Brain distribution of *Gpr149* mRNA in a C57Bl6 male mouse.** A series of digital scans summarizing brain sites with high *Gpr149* mRNA expression. Fluorescent RNAscope signals for *Gpr149* mRNA (opal 570 converted to black and white) were captured using a Zeiss Axioscan.Z1. A complete list of brain sites with *Gpr149* expression and abbreviations can be found in Table 1.

(Fig. 5H). This may result from developmental expression of Gpr149-Cre. Overall, this novel *Gpr149*-Cre line is both sensitive and specific.

The distribution of tdTomato was further investigated in peripheral tissues. Many tdTomato-positive neurons are observed in the sensory ganglia (Figs. 6A and 6B). In the

**Table 1 Qualitative estimates of Gpr149 mRNA expression in the mouse brain as assessed by fluorescent RNAscope.**

| Brain structure(s) | Relative RNAscope signals density |
|---|---|
| Dorsal horn of the spinal cord | ++++ |
| Dorsal tegmental nucleus (DTg) | ++++ |
| Interpeduncular nucleus rostral nucleus (IPR) | ++++ |
| Island of Calleja (ICj) | ++++ |
| Laterodorsal tegmental nucleus (LDTg) | ++++ |
| Olfactory tubercle (Tu) | ++++ |
| Posterodorsal tegmental nucleus (PDTg) | ++++ |
| Sphenoid nucleus (Sph) | ++++ |
| Ventral pallidum (VP) | ++++ |
| Ventromedial hypothalamus (VMH) | ++++ |
| 12N | ++ |
| 5N | ++ |
| 7N | ++ |
| AcbSh | ++ |
| Bar | ++ |
| DIC | ++ |
| EIC | ++ |
| Edinger-westphal nucleus (EW) | ++ |
| Globus pallidus (GP) | ++ |
| Intercalated nuclei of the amygdala (I) | ++ |
| Interfascicular nucleus (IF) | ++ |
| Interpeduncular nucleus caudal subnucleus (IPC) | ++ |
| Interpeduncular nucleus lateral subnucleus (IPL) | ++ |
| Lateral periaqueductal gray (LPAG) | ++ |
| Lateral parabrachial nucleus dorsal part (LPBD) | ++ |
| Lateral septal nucleus intermediate part (LSI) | ++ |
| Lateral stripe of the striatum (LSS) | ++ |
| Medial amygdaloid nucleus anterior part (MeA) | ++ |
| Medial amygdaloid nucleus posteroventral part (MePV) | ++ |
| Microcellular tegmental nucleus (MiTg) | ++ |
| Navicular postolfactory nucleus (Nv) | ++ |
| Optic nerve layer of the superior colliculus (Op) | ++ |
| Peduncular part of the lateral hypothalamus (PLH) | ++ |
| Pre-edinger-westphal nucleus (PrEW) | ++ |
| Retrochiasmatic area lateral part (RchL) | ++ |
| Rostral linear nucleus (RLi) | ++ |
| Retromammillary nucleus lateral part (RML) | ++ |
| Retromammillary nucleus medial part (RMM) | ++ |
| Accumbens nucleus, core (AcbC) | + |
| Anterior cortical amygdaloid area (ACo) | + |
| Anterior hypothalamic area, central part (AHC) | + |

| Brain structure(s) | Relative RNAscope signals density |
|---|---|
| Area postrema (AP) | + |
| Arcuate nucleus of the hypothalamus (Arc) | + |
| Basolateral amygdaloid nucleus, anterior part (BLA) | + |
| Basolateral amygdaloid nucleus, ventral part (BLV) | + |
| Basomedial amygdaloid nucleus, anterior part (BMA) | + |
| Central medial thalamic nucleus (CM) | + |
| Caudate putamen (Cpu) | + |
| Dorsal lateral geniculate nucleus (DLG) | + |
| Dorsomedial tegmental area (DMTg) | + |
| Intermediate white layer of the superior colliculus (InWh) | + |
| Lateral parabrachial nucleus (LPB) | + |
| Lateral parabrachial nucleus, superior part (LPBS) | + |
| Lateral reticular nucleus (lRt) | + |
| Medial geniculate nucleus, ventral part (MGV) | + |
| Medial habenular nucleus (MHb) | + |
| Medial mammillary nucleus, median part (MnM) | + |
| Median raphe nucleus (MnR) | + |
| Medial preoptic area (MPA) | + |
| Periacqueductal gray (PAG) | + |
| Posterior hypothalamus (PH) | + |
| Posteromedial cortical amygdaloid area (PMCo) | + |
| Premammillary nucleus, dorsal part (PMD) | + |
| Retrochasmatic area (Rch) | + |
| Raphe magnus nucleus (RMg) | + |
| Raphe pallidus nucleus (Rpa) | + |
| Retrorubral field (RRF) | + |
| Reticular nucleus (Rt) | + |
| Spinal trigeminal tract (sp5) | + |
| Subbrachial nucleus (sub) | + |
| Superficial gray layer of the superior colliculus (SuG) | + |
| Ventrolateral thalamic nucleus (VL) | + |
| Vascular organ of the lamina terminalis (VOLT) | + |
| Ventral posterolateral thalamic nucleus (VPL) | + |
| Ventral posteromedial thalamic nucleus (VPM) | + |
| Ventral tergmental nucleus (VTg) | + |
| Ventral tuberomammillary nucleus (VTM) | + |
| Zona incerta (ZI) | + |
| Ambiguous nucleus (Amb) | +/− |
| Field CA3/2 of hippocampus (CA3/2) | +/− |
| Dentate gyrus (DG) | +/− |
| Dorsal motor nucleus of the vagus (DMX) | +/− |

| Brain structure(s) | Relative RNAscope signals density |
|---|---|
| Dorsal raphe nucleus (DR) | +/− |
| Lateral habenular nucleus (LHb) | +/− |
| Piriform cortex (Pir) | +/− |
| Solitary nucleus (Sol) | +/− |

**Note:**
A total of three different brains were visually inspected to produce the above estimates. The following density scale of signal strength was used: ++++, highest density; +++, high density; ++, moderate density; +, low density; +/−, inconsistent signals. Brain structures were ordered by signals density (highest to lowest) and alphabetic order. The nomenclature is based on the Mouse Brain in Stereotaxic Coordinates (3rd edition) by Franklin and Paxinos.

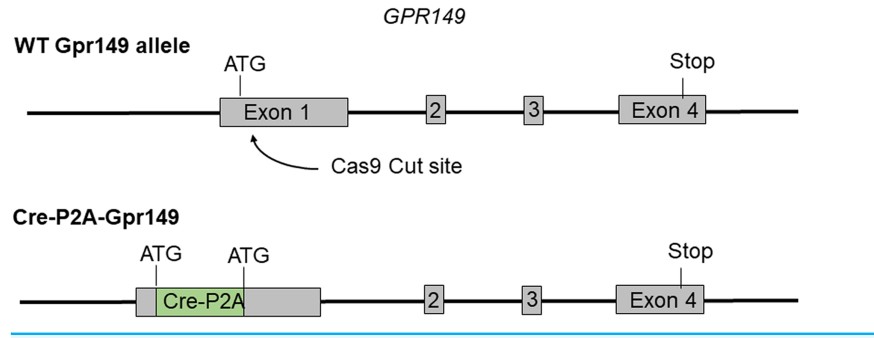

**Figure 3 Schematic representation of the transgenes used to generate Gpr149-Cre mice.** Using CRISPR/Cas9 a 1,119 bp Cre-P2A sequence was knocked into the start codon of *Gpr149* exon 1.

superior cervical ganglion, only tomato axons rather than cell bodies are observed (Fig. 6C).

The enteric nervous system is also labeled (Fig. 6C). In addition to enteric neurons, unidentified cells resembling vascular cells and interstitial muscular cells are seen in the gastrointestinal tract (Figs. 6D–6G). In agreement with our qPCR data, the pituitary gland anterior lobe also contains a dense network of cells resembling endocrine cells (Fig. 7A). The pancreas only contains a few axons, presumably originating from sensory neurons (Figs. 7B and 7C). Many other organs present sparsely distributed cells of unknown identity, including the spleen (Fig. 7D), liver (Fig. 7E), kidney (Fig. 7F), and heart (Fig. 7G). Because the latter cells are frequently around blood vessels, we deduced they may be (peri) vascular cells.

## Novel Gpr149 knockout mice partially resist diet-induced obesity

Using CRISPR-Cas9, we generated a global knockout allele with mice lacking *Gpr149* exon 1 (Fig. 8A). *Gpr149*$^{-/-}$ mice were viable and born without overt phenotypes at expected mendelian ratios (Table 2). Loss of hypothalamic expression for *Gpr149* expression was confirmed in the knockout model by qPCR (Fig. 8B). Raw qPCR data and experimental details are included in Fig. S2. In addition, the chromogenic (red) RNAscope® system allowed us to further verify the loss of *Gpr149* expression in three representative tissues with high *Gpr149* expression, including the VMH (Fig. 9A), nodose ganglion (Figs. 9C and

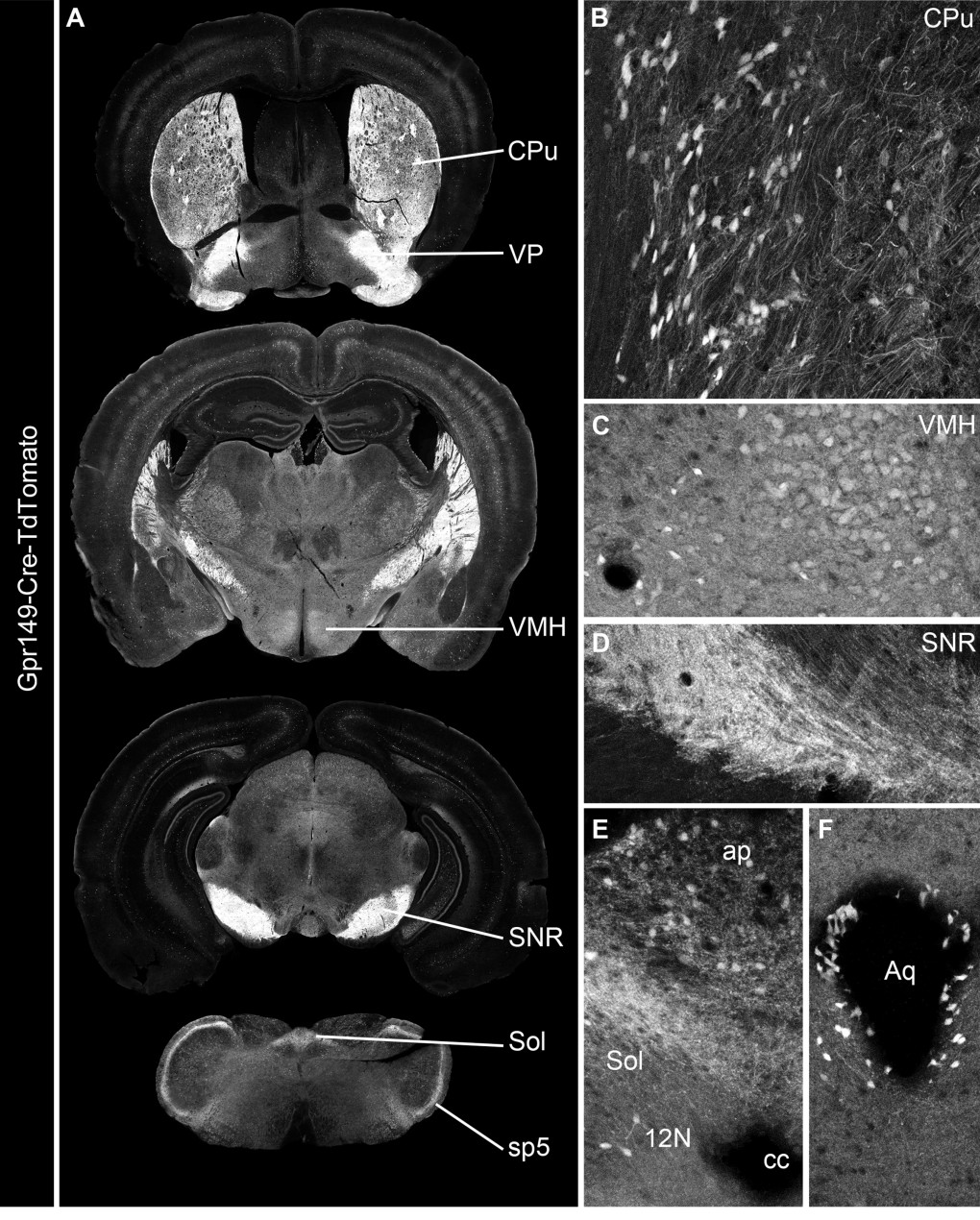

**Figure 4 Generation of the novel Gpr149-Cre-tdTomato mouse line.** (A) A series of digital scans representative of the brain of Gpr149-Cre-tdTomato male mice. Native tdTomato fluorescence (converted to black and white) was captured using a Zeiss Axioscan.Z1. Intense fluorescence is observed in fiber tracts and brain nuclei. (B and C) High-magnification confocal images of representative brain regions in Gpr149-Cre-tdTomato male mice. Cell bodies resembling neurons, often intermingled with axons, are labeled in numerous sites previously reported to express *Gpr149*, including the Cpu and VMH. (D) Several areas contain mostly axons including, most notably, the SNR, which is heavily enriched in tomato fibers. (E) The dorsovagal complex contains both sparse neurons and fiber tracts presumably originating from the vagal ganglia. (F) The epithelial lining of the cerebral aqueduct often contains non-neuronal cells. This is the only brain region containing seemingly non-neuronal cells. Abbreviations can be found in Table 1.               

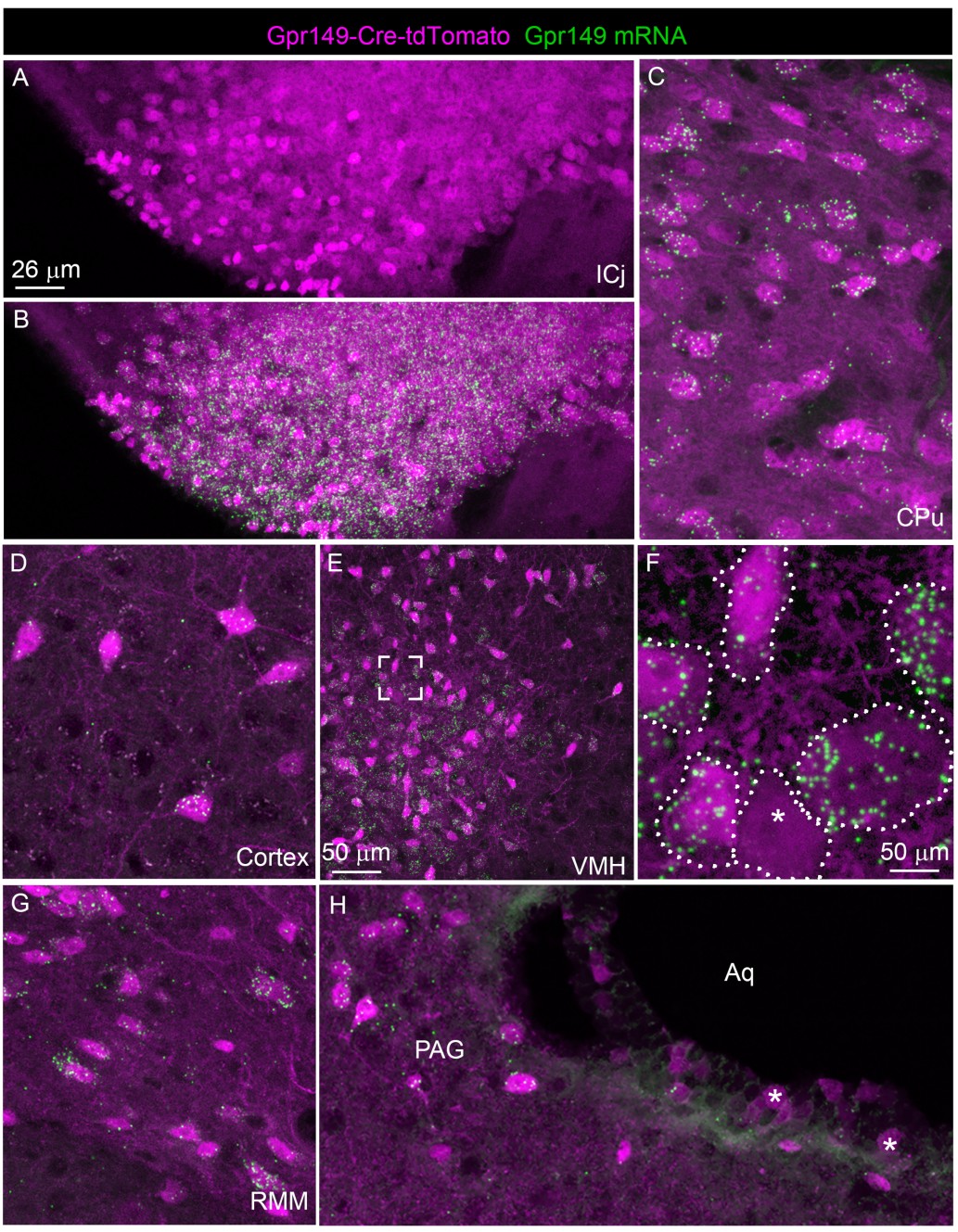

**Figure 5 Validation of the Gpr149-Cre-tdTomato mouse line.** (A–H) RNAscope analysis for *Gpr149* (green dots) was performed on the brain of Gpr149-Cre-tdTomato mice (magenta cells). Brain regions with *Gpr149* signals without tdTomato-positive cell bodies are observed. Moreover, strong signals are seen in the vast majority of tdTomato-labeled cell bodies across the brain. Tomato cells devoid of *Gpr149* signals are only seen occasionally in most brain regions (indicated by *). The only exception is the lining of the cerebral aqueduct, where non-neuronal cells express low levels or no *Gpr149* signals. Scale bar in A applies to B, C, D, G, and H.

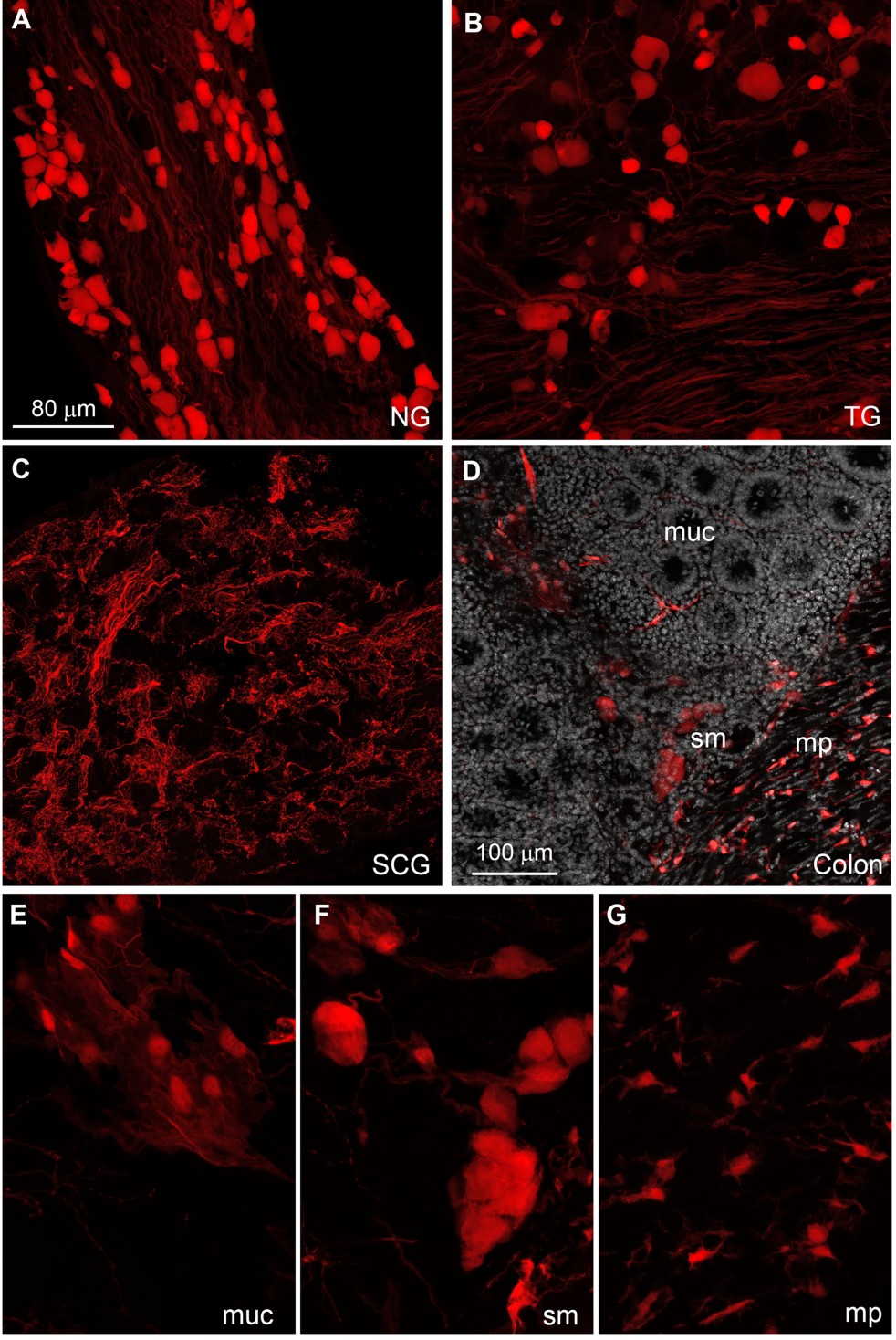

**Figure 6 Distribution of Gpr149-Cre-tomato cells in the peripheral nervous system.** (A–G) Representative confocal images of Tomato-positive cells in the peripheral nervous system. (A and B) All sensory ganglia contain a large subset of tdTomato-positive neurons. (C) In contrast, sympathetic motor ganglia contain only tdTomato-positive axonal fibers without identifiable cell bodies. (D–G) The entire gastrointestinal tract contain fluorescence-labeled cells across both the myenteric and mucosal layers. (E) In the mucosa, no epithelial or immune cells are readily observed, but patches of large cells resembling (peri)vascular cells are presented. (F) Enteric neurons are clearly seen in the myenteric and submucocal

**Figure 6 (continued)**
layers within the small ganglia. (G) The muscle layers also contain small unidentified cells intercalated between muscle fibers. Their morphology and distribution pattern resembled those of interstitial Cajal cells. Abbreviations: NG, nodose ganglion; mp, myenteric plexus muc, mucosa; SCG, superior cervical ganglion; sm, submucosa; TG, trigeminal ganglion.     

9D), and pituitary gland (Figs. 9G and 9H). Gpr149 transcripts were virtually undetectable in the same tissues from the $Gpr149^{-/-}$ mouse (Figs. 9B, 9E, 9F, 9I and 9J). Additional histological validation on the trigeminal ganglion and spinal cord is available in Fig. S3. While there is little reason to believe that Gpr149 would not be absent from tissues of knockout mice, it is important to note that our analysis solely focused on tissues with moderate and high expression, which is a caveat that should be acknowledged.

Next, a preliminary metabolic profile analysis of the $Gpr149^{-/-}$ mouse was performed. (Fig. 10). When fed a standard chow diet, there is a trend toward a lower weight gain rate in the $Gpr149^{-/-}$ mouse (Fig. 10A). When switched to HFD (60% fat), diet-induced obesity was significantly less pronounced in $Gpr149^{-/-}$ mice (Fig. 10A). $Gpr149^{+/-}$ mice show an intermediate phenotype. This is accompanied by a reduction in fat mass, as determined by NMR (Fig. 10B), independent of any differences in body length (Fig. 10C). Interestingly, organ weights were not statistically different between genotypes (Fig. 10D). Over a period of 4 h, separate cohorts of fasted refed mice of the three different genotypes ate the same amount of standard chow (Fig. 10E). Under conditions of *ad libitum* food access, the three different genotypes also ate the same amount of HFD on a daily basis (Fig. 10F). Glucose tolerance tests in a separate cohort fed on standard chow show no difference between genotypes (Fig. 10G). However, both $Gpr149^{-/-}$ and $Gpr149^{+/-}$ mice had greater sensitivity to insulin than the control mice (Fig. 10H). The above data indicate that GPR149 is involved in energy balance and glucose homeostasis. Raw physiological data, detailed statistical results, and diets composition are included in Fig. S4.

## DISCUSSION

Using qPCR and ISH, we found high *Gpr149* expression in brain regions controlling energy expenditure, food intake, and glucose homeostasis. This included the VMH, a region well known to be required for energy balance and glucose homeostasis regulation (*Dhillon et al., 2006*; *Castorena et al., 2021*). This finding is in good agreement with those from other studies (*Ehrlich et al., 2018*; *Affinati et al., 2021*). We also found that *Gpr149* is particularly enriched in striatal hedonic circuits driving food intake. Likewise, we detected very high levels of *Gpr149* in the islands of Calleja, which are situated in rewarding nuclei of the ventral striatum on the border of the nucleus accumbens (*Zhang et al., 2021*). These regions receive dense innervation from the dopaminergic system and have high levels of dopamine receptor (*Zhang et al., 2021*). Classic lesion and genetic studies of this brain region have implicated it in the rewarding effects of many substances including highly palatable diets (*Schwartz et al., 1998*). The island Calleja has also recently been linked to grooming behaviors (*Zhang et al., 2021*). Although no obvious feeding or grooming

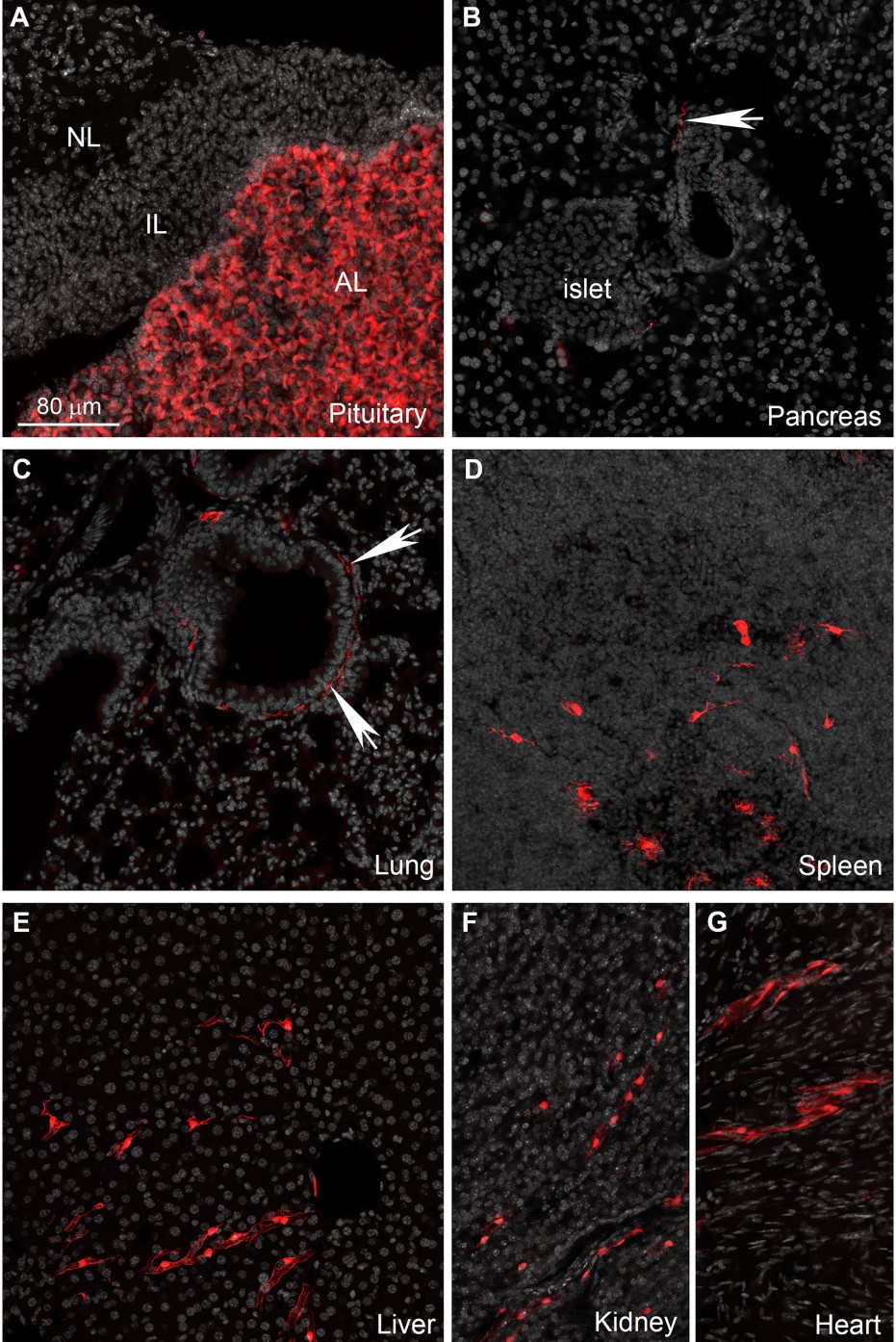

**Figure 7  Distribution of Gpr149-cre-Tomato cells in different organs.** (A–G) Representative confocal images of tdTomato-positive cells in peripheral organs. Tissues were counterstained with DAPI (grey). (A) The pituitary gland peripheral is the organ with the most abundant tdTomato fluorescence. The anterior lobe contains a dense network of endocrine cells positive for tdTomato. (B and C) Sparse axonal fibers of presumptive autonomic origin are discerned in many organs, including the lungs and heart (white arrows). (D) In addition, small clusters of unidentified cells are observed in the spleen. Their morphology was reminiscent dendritic cells rather than of lymphocytes. (E, F and G) Lastly, most tissues also contain sparse clusters of cells with the distribution and shape of perivascular cells. Abbreviations: AL, anterior lobe; IL, intermediate lobe; NL, neural lobe.

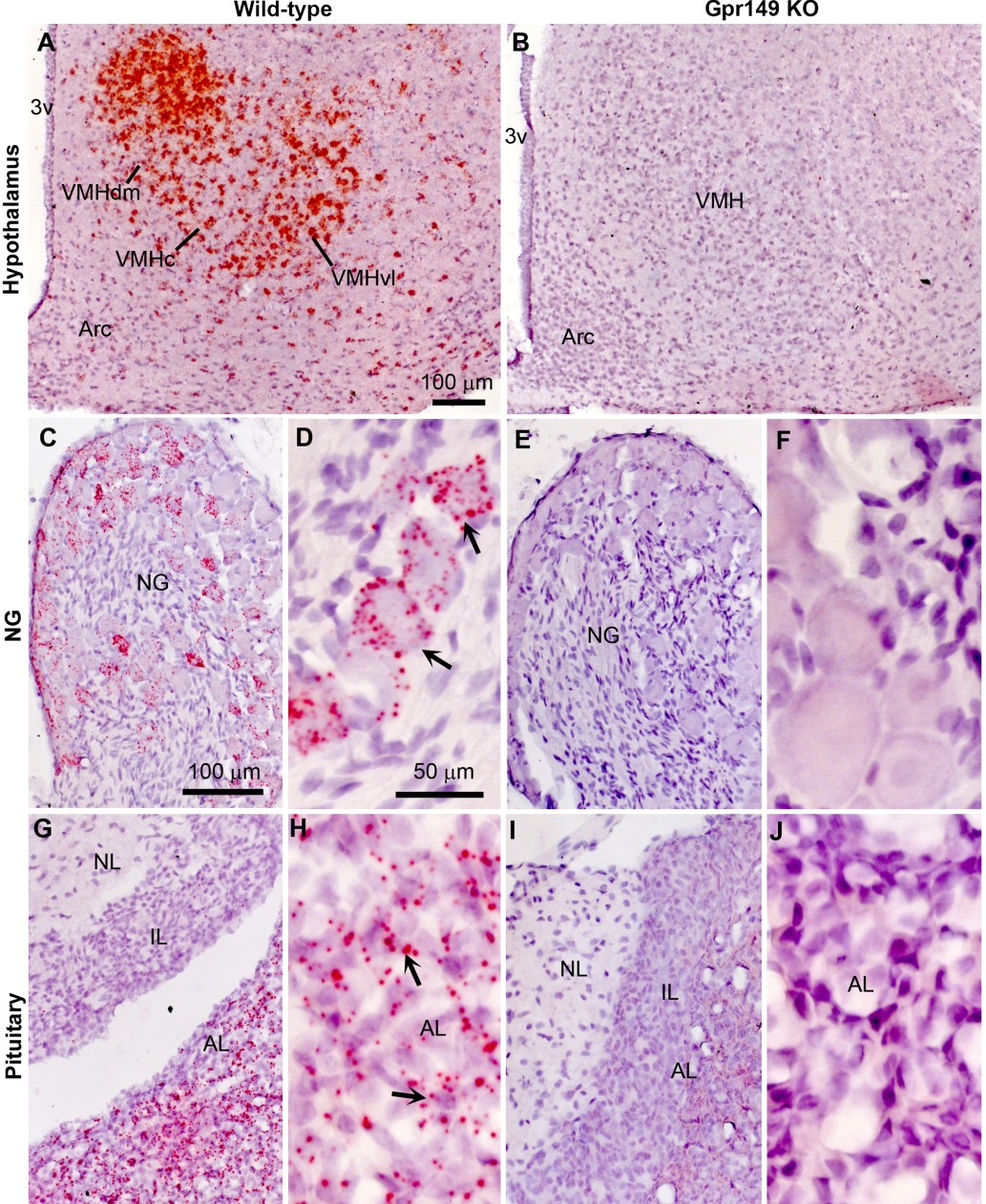

**Figure 8 Histological validation of the *Gpr149*⁻/⁻ mouse line.** Chromogenic RNAscope ISH was used to assess *Gpr149* expression (red dots) in wildtype *vs Gpr149*⁻/⁻ mice. Tissues were counterstained with hematoxylin and imaged with bright-field microscopy. In the wild type mice, robust *Gpr149* signals are observed in the entire VMH (A) the nodose ganglion (C and D), and the anterior lobe of the pituitary gland (G and H). Representative cells with high levels of *Gpr149* expression are indicated with black arrows. In the *Gpr149*⁻/⁻ mouse, the same tissues are completely devoid of signals (B, E, F, I and J). Abbreviation: AL, anterior lobe; Arc, arcuate nucleus; NG, nodose ganglion; IL, intermediate lobe; NL, neural lobe; VMHc, ventromedial hypothalamus compact part; VMHdl, ventromedial hypothalamus dorsolateral part; VMHdm, ventromedial hypothalamus dorsomedial part. Scale bar in A applies to B. Scale bar in C applies to I, G, I. Scale bar in D, F, H, J.

Table 2 **Gpr149 knockout mice are born at a normal mendelian ratio.**

| Gpr149 genotype | Expected no. (%) | Observed no. (%) |
|---|---|---|
| Gpr149$^{+/+}$ | 11 (25%) | 6 (14%) |
| Gpr149$^{+/-}$ | 22 (50%) | 29 (66%) |
| Gpr149$^{-/-}$ | 11 (25%) | 9 (20%) |
| Total | 44 (100%) | 44 (100%) |
| Chi test $p$ value | 0.946230961 | |

**Note:**
The number of mice of various genotypes (and percentage) is shown. The expected number of mice was calculated based on the expected mendelian ratio of 1:2:1.

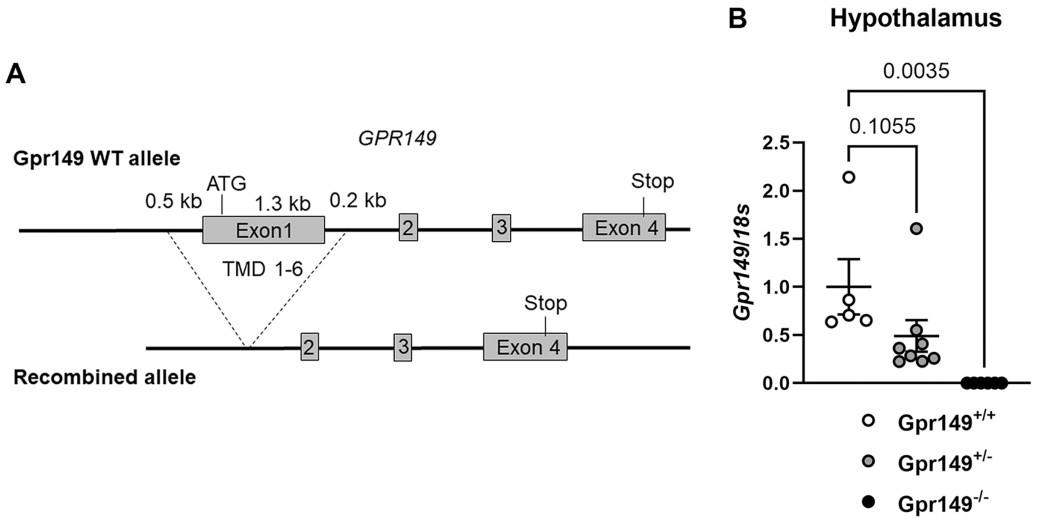

**Figure 9 Generation of the *Gpr149*$^{-/-}$ mouse line.** (A) Schematic representation of the transgenes used to generate the *Gpr149*$^{-/-}$ mice. Using CRISPR/Cas9, we used two guides to target Exon1 which encodes the start codon and the first 6 transmembrane domains (TMD1-6). (B) QPCR for *Gpr149* expression in the hypothalamus from wildtype *vs Gpr149*$^{+/-}$ and *Gpr149*$^{-/-}$ mice ($N = 5$–8 mice). Data were analyzed using a One-way ANOVA followed by a Dunnett *post-hoc* comparison. The exact p-value is listed each bar graph.

phenotypes were detected in *Gpr149*$^{-/-}$ animals, it cannot be ruled out that *Gpr149*$^{-/-}$ may display altered reward and feeding behaviors in more sophisticated tests. Pituitary endocrine cells including prolactin cells also express *Gpr149*, an observation consistent with the previously established link between GPR149 and reproductive functions (*Edson, Lin & Matzuk, 2010*). Gpr149 expression in the periphery was found to be low but not completely absent. To the best of our knowledge, the role of Gpr149 signaling in peripheral tissues remains entirely unknown. According to our tdTomato mapping data, Gpr149-positive cells comprise a small subset of immune and/or vascular cells. Further investigations are necessary to explore the potential implications of Gpr149 in vascular biology and immunity.

In summary, GPR149-bearing neurons and endocrine cells may serve as links between changing levels of peripheral metabolic cues and CNS pathways controlling energy balance, glucose homeostasis, and reproduction. However, given how widely *Gpr149* is

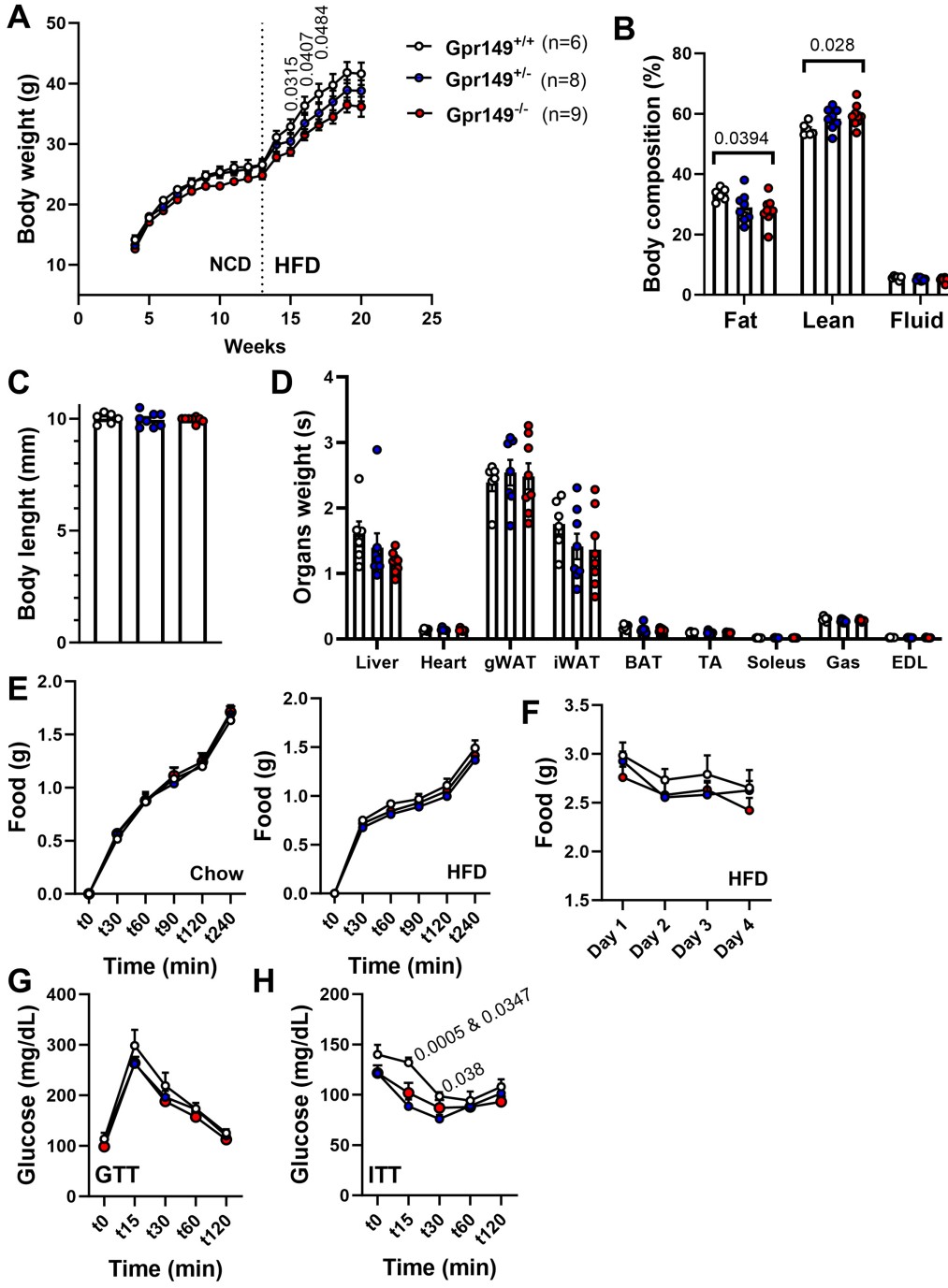

**Figure 10 Preliminary metabolic profiling of the *Gpr149*⁻/⁻ mouse line.** (A) Cumulative weekly body weights of WT, *Gpr149*⁺/⁻, and *Gpr149*⁻/⁻ mice fed on normal chow and switched to a high fat diet at week 13. Data were analyzed using a two-way ANOVA followed by a Dunnett *post-hoc* comparison. The exact *p*-value is listed above a timepoint (when $p < 0.05$ between WT and *Gpr149*⁻/⁻). (B) Body composition assessed by NMR at week 20. Data were analyzed using ANOVA. The exact *p*-value is listed above a bar graph (when $p < 0.05$ between WT and *Gpr149*⁻/⁻) and detailed statistical results are in Fig. S3. (C) Body length at week 20. (D) Weight of organs collected at week 20. Data were analyzed using a one-way ANOVA, but not differences between genotypes were found. (E) Cumulative feeding of chow and HFD of cohorts of WT, *Gpr149*⁺/⁻, and *Gpr149*⁻/⁻ mice submitted to a fasting-refeeding challenge. (F) Daily food intake of WT, *Gpr149*⁺/⁻, and *Gpr149*⁻/⁻ mice with unrestricted access of HFD. (G) Glucose

**Figure 10 (continued)**
tolerance test (GTT) and (H) insulin tolerance test (ITT) results from cohorts of WT, $Gpr149^{+/-}$ and $Gpr149^{-/-}$ mice fed on chow diet. Data were analyzed using a Two-way ANOVA followed by a Dunnett *post-hoc* comparison. The exact p-value is listed above a timepoint when $p < 0.05$ (first value between WT and $Gpr149^{+/-}$; second value between WT and $Gpr149^{-/-}$). All data are presented as mean ± SEM. Detailed statistical results are in Fig. S3. Abbreviations: NCD, normal chow diet; HFD, high-fat diet; gWAT, gonadal white adipose tissue; iWAT, inguinal white adipose tissue; BAT, brown adipose tissue; TA, tibialis anterior; Gas, gastrocnemius; EDL, extensor digitorum longus.

expressed in the brain, we propose that the importance of GPR149 extends beyond the field of obesity and diabetes research. For instance, GPR149 signaling is likely involved in nociception given the expression of *Gpr149* in the primary sensory areas, periaqueductal grey and thalamus. In further support of this view, Gpr149 expression occurred in the dorsal horn and trigeminal ganglia which are important relays in the ascending pathway for nociception. Moreover, many Gpr149-expressing forebrain and hypothalamic sites have been implicated in modulating nociception (*Luo et al., 2023*; *Harris & Peng, 2020*; *Borszcz, 2006*). While our data indicate a physiological role of GPR149 signaling in the context of diet-induced obesity and glucose homeostasis, additional studies are needed to understand the exact mechanisms linking GPR149 and the neural control of energy balance in both males and females. Specifically, there is a need for further metabolic profiling in animals during the later stages of life. Moreover, we hypothesize that deleting Gpr149 in adult mice, rather than during their developmental stages, may lead to a more pronounced increase in body weight.

Pharmaceutical research has focused for many years on GPCRs in the search for anti-obesity and diabetes drugs (*Kievit et al., 2013*; *Sanchez-Garrido et al., 2017*). The value of GPCRs-directed drugs in the field of metabolic research is proven (*Burke et al., 2017*; *Mul et al., 2013*), and they are effective treatments for obesity and type 2 diabetes. Nonetheless, there are few currently approved GPCR-directed drugs, most present side effects, and are often less effective than bariatric surgery (*Kim, Seeley & Sandoval, 2018*). Thus, more effective drug targets are needed in the field of metabolic research.

Our current findings underscore the potential of GPR149 as a promising drug target for the treatment of metabolic diseases. Furthermore, our research provides investigators interested in GPCRs novel genetic tools that may help explore the role of GPR149 *in vivo*. Our Cre line may be a useful means to manipulate specific populations of Grp149-expressing cells with opto- and chemo-genetic tools. It could also be used for the purpose of tracing, patch clamp, and cell sorting studies. currently, GPR149 remains an orphan receptor without known ligand(s) and downstream signaling pathways. Thus, our novel knockout mouse may help identify modulators of GPR149 signaling and validate potential antibodies against GPR149.

## ACKNOWLEDGEMENTS

We would like to also thank the Whole Brain Microscopy Facility at UT Southwestern Medical Center (RRID:SCR_017949) for assistance with slide scanning (Dr. Denise

Ramirez). We also would like to thank Chelsea Limboy (UTSW) for her technical help. A preliminary version of this study was previously shared with the scientific community during the NIH Illuminating the Druggable Genome online meeting (2022) and the ObesityWeek meeting (2023).

### Funding
This work was supported by the National Insitute of Health R03TR003655-01 (Neuronal Gpr149 and energy balance), P01DK119130 (CNS mechanisms linking exercise training with energy balance and metabolism, Core C), and 1P30Dk127984-01A1 (NORC to UTSW). The funders had no role in study design, data collection and analysis, decision to publish, or preparation of the manuscript.

### Grant Disclosures
The following grant information was disclosed by the authors:
National Insitute of Health: R03TR003655-01, P01DK119130 and 1P30Dk127984-01A1.

### Competing Interests
The authors declare that they have no competing interests.

### Author Contributions

- Steven Wyler conceived and designed the experiments, performed the experiments, analyzed the data, prepared figures and/or tables, authored or reviewed drafts of the article, and approved the final draft.
- Surbhi performed the experiments, authored or reviewed drafts of the article, and approved the final draft.
- Newton Cao performed the experiments, authored or reviewed drafts of the article, and approved the final draft.
- Warda Merchant performed the experiments, authored or reviewed drafts of the article, and approved the final draft.
- Angie Bookout performed the experiments, authored or reviewed drafts of the article, and approved the final draft.
- Laurent Gautron conceived and designed the experiments, analyzed the data, prepared figures and/or tables, authored or reviewed drafts of the article, and approved the final draft.

### Animal Ethics
The following information was supplied relating to ethical approvals (*i.e.*, approving body and any reference numbers):

Our animal studies were reviewed and approved by the UT Southwestern Medical Center IACUC.
## Data Availability

The raw data for qPCR assays and physiological studies and the sequences of our guid probes are available in the Supplemental Files.

## Supplemental Information

Supplemental information for this article can be found online at http://dx.doi.org/10.7717/peerj.16739#supplemental-information.

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
