# Peer review of "Gpr149 is involved in energy homeostasis in the male mouse"

_PeerJ, doi:10.7717/peerj.16739_

## Round 0.1 · original submission · Minor Revisions

All three reviewers are largely positive, but all have some suggestions for improvement which are largely straightforward and reflect mainly stylistic changes or points of clarification.

Reviewer 1 ·

Basic reporting

This manuscript is well written and provide new insights into the potential physiological role of the orphan GPCR, GPR149. Overall, the manuscript should be accepted with either minor revisions or additional comments from the authors.

Experimental design

Correction on line 155. t-test is not described in detail (paired vs unpaired). Were there any corrections applied to the t-test?

Validity of the findings

Line 167- Is there qPCR data to distinguish GPR149 expression in subpopulations of the hypothalamus?
Line 221-222- Can the authors provide additional comments on the low expressions in peripheral organs in the discussion section?
Line 226-227- Was the expression of GPR149 at other regions of the brain/ peripheral tissues also knocked out? This was expanded in the chromogenic RNAscope histological study, but was histological characterisation not performed on other tissues as well? For example, did expression drop in certain tissues associated with nociception (e.g. TG/ dorsal horn)?
Line 234-235- It would be ideal to see a separate cohort continuing the the standard chow to see if weight loss also occurs with standard chow. This would give more insight if GPR149 alters appetite/metabolism in baseline conditions. This is something for the authors to consider commenting in the discussion.
Line 235-239- The reduction in general mass and fat mass seen in NMR but not in organs or visceral and subcutaneous fat is interesting. Are there any data indicative of changes in water intake or perhaps bone densities. Perhaps the authors can comment further on this. Additionally, body length was captured at week 20, but body length captured prior to HFD may help elucidate if development of mice is affected by GPR149 KO. Perhaps authors can comment on this as well.
Line 239-240- It is rather unfortunate that there was no difference in the re-feed experiment, but perhaps 4 hours may have been too short of a time period to identify differences. Were there any distinction between cohorts for meal size and feeding length and frequency within this 4 hours? This is a particularly important distinction given that the authors suggest GPR149 expression in brain region is associated with hunger/ satiety and food reward regulation. It is recommended that authors present data or perform further experiments characterizing food intake in detail. Although authors have commented on the possibility of altered feeding behaviours in the discussion, this data would highly support the potential utility of GPR149 KO mice in future metabolic studies.
Line 241-243- Both ITT and GTT were performed on mice cohorts fed standard chow. Authors may want to consider performing ITT and GTT in cohorts fed HFD to better inform if GPR149 KO has protective glucose homeostasis properties.

Additional comments

Line-266-267- It is likely as per the authors suggestion that GPR149 plays a role in descending pathways for nociception. But authors should also acknowledge GPR149 expression in specific sites such as the dorsal horn and TG in this sentence which are also heavily associated with the ascending pathway for nociception. Authors may find it very useful to include in the discussion parallels between GPR149 expression and possible physiological roles to GPCRs known to have similar expression profiles and already established to be involved in both nociception and metabolism, i.e. CGRP and amylin receptors. This would further strengthen authors hypothesis.

Reviewer 2 ·

Basic reporting

no comment

Experimental design

This manuscript presents interesting findings from rigorously designed experiments aimed at determining the distribution and function of an orphan GPCR, Gpr149, in mice. The authors employed high technical standards to map Gpr149 across tissues and developed a novel Gpr149-Cre mouse line, validated for future studies to uncover the underlying neuroanatomical basis of Gpr149 pathways. Although a Gpr149 knockout mouse line had previously been generated and validated, the authors' effort in generating an additional line is admirable, although the reason for doing so is not entirely clear. The metabolic phenotyping conducted is somewhat superficial, but the reported parameters have sufficient N/group, and the conclusion is strongly supported by the results. However, as the authors stated, more comprehensive metabolic phenotyping is necessary to understand the exact mechanisms linking Gpr149 and metabolic homeostasis in both males and females.

Validity of the findings

The results being reported are solid and statistically sound, and the conclusions are reasonably stated and supported by well-controlled experiments.

Reviewer 3 ·

Basic reporting

-The manuscript is for the most part clearly written with an appropriate introduction and background.

-Figures are mostly well presented, although the qPCR figures needs some clarification on the units presented on the y axis (I gather this must be a % but it is not stated).

-In Figure 1 it is not clear to me why error bars have not been included. When statements have been made in the text about differences in levels of expression, are these conclusions statistically supported?

Experimental design

-The objective of this work was clearly explained, it was not hypothesis driven.

- The experiments present address the objectives set out.

- Methods are generally clear, however more detail should be included to explain how qPCR data has been quantified (what references are used) in the methods.

- It is also important to note that the authors have not included any controls for their imaging (either the ISH or TdTomato). In both cases including negative controls to confirm signals are specific would be helpful. That said, the fact that similar patterns of GPR149 expression are mostly seen between these two approaches suggests the signal is likely specific. This could be discussed more in the manuscript.

Validity of the findings

-Other than the previous note about the limited controls presented for the imaging work, the findings appear to be robust and valid.

-Conclusions are clear and underlying data have been included.

Additional comments

In line 155 the text includes: "...or a t-Test (?????)." which is I guess a typo.

---

## Round 0.2 · accepted · Accept

Thank you for carefully addressing all the points. I am delighted to indicate acceptance.